# Parametric matrix models

Patrick Cook [1,2,5], Danny Jammooa [1,2,5], Morten Hjorth-Jensen [1,2,3], Daniel D. Lee [4] & Dean Lee [1,2] ✉

We present a general class of machine learning algorithms called parametric matrix models. In contrast with most existing machine learning models that imitate the biology of neurons, parametric matrix models use matrix equations that emulate physical systems. Similar to how physics problems are usually solved, parametric matrix models learn the governing equations that lead to the desired outputs. Parametric matrix models can be efficiently trained from empirical data, and the equations may use algebraic, differential, or integral relations. While originally designed for scientific computing, we prove that parametric matrix models are universal function approximators that can be applied to general machine learning problems. After introducing the underlying theory, we apply parametric matrix models to a series of different challenges that show their performance for a wide range of problems. For all the challenges tested here, parametric matrix models produce accurate results within an efficient and interpretable computational framework that allows for input feature extrapolation.

One of the first steps in solving any physics problem is identifying the governing equations. While the solutions to those equations may exhibit highly complex phenomena, the equations themselves have a simple and logical structure. The structure of the underlying equations leads to important and nontrivial constraints on the analytic properties of the solutions. Some well-known examples include symmetries, conserved quantities, causality, and analyticity. Unfortunately, these constraints are not always reproduced by machine learning algorithms, leading to inefficiencies and limited accuracy for scientific computing applications. Many constraints can be guaranteed by the architecture of specifically designed neural networks, such is the case with equivariant neural networks. However, it is not straightforward to construct a neural network that conforms to more complicated physical constraints and as such current physics-inspired and physics-informed machine learning approaches aim to constrain the solutions by penalizing violations of the underlying equations[1–3], but this does not guarantee exact adherence. Furthermore, the results of many modern deep learning methods suffer from a lack of interpretability. To address these issues, in this work we introduce a class of machine learning algorithms called parametric matrix models. Parametric matrix models (PMMs) work by replacing operators in the known or

supposed governing equations with trainable, parametrized ones. Parametric matrix models take the additional step of applying the principles of model order reduction and reduced basis methods[4,5] to find efficient approximate matrix equations with finite dimensions. Such equations are guaranteed to exist and can be constructed, in theory, via methods such as the proper orthogonal decomposition[6].

While most machine learning methods optimize some prescribed functional form for the output functions[2,7–12], PMMs belong to a class of machine learning methods based on implicit functions[13–18]. We define a PMM as a set of matrix equations with unknown parameters that are optimized to express the desired outputs as implicit functions of the input features. There are many possible choices for the matrix equations. When the form of the governing equations is known or supposed, as is the case for data from physical systems, the form of the PMM will be analogous to these equations. This is what lends many PMMs their interpretability.

In this work, we define a basic PMM form composed of primary matrices, $P_j$, and secondary matrices, $S_k$. All of the matrix elements of the primary and secondary matrices are analytic functions of the set of input features, $\{c_l\}$. For dynamical systems, one of the input features would be time. We consider functional forms for the primary and

[1]Facility for Rare Isotope Beams, Michigan State University, East Lansing, MI, USA. [2]Department of Physics and Astronomy, Michigan State University, East Lansing, MI, USA. [3]Department of Physics and Center for Computing in Science Education, University of Oslo, Oslo, Norway. [4]Department of Electrical and Computer Engineering, Cornell Tech, New York, NY, USA. [5]These authors contributed equally: Patrick Cook, Danny Jammooa. ✉e-mail: leed@frib.msu.edu

second matrix elements that are preserved under unitary transformations of the matrices. The primary matrices are, in this work, square matrices that are either Hermitian matrices or unitary matrices, and their corresponding normalized eigenvectors, $\mathbf{v}_j^{(i)}$, are used to form bilinears with secondary matrices of appropriate row and column dimensions to form scalar outputs of the form, $\mathbf{v}_j^{(i)\dagger} S_k \mathbf{v}_{j'}^{(i')}$. When $i = i'$ and $j = j'$, the resulting output is the expectation value of $S_k$ associated with eigenvector $\mathbf{v}_j^{(i)}$. When $i \neq i'$ or $j \neq j'$, the output is the transition amplitude induced by $S_k$ between eigenvectors $\mathbf{v}_j^{(i)}$ and $\mathbf{v}_{j'}^{(i')}$. Since the transition amplitudes carry an arbitrary complex phase, we work with transition probabilities composed of squared absolute values of the amplitudes. Depending on the structure of the specific problem the PMM is being applied to, either the primary or the secondary matrices may be completely omitted—as is the case with unitary time evolution and eigenvalue emulation respectively. We note the close analogy between outputs of PMMs of this basic form and observables in quantum mechanics. This has the practical benefit that gradients with respect to any unknown parameters can be computed efficiently using first-order perturbation theory[19,20]. This design feature is very useful for efficient parameter optimization.

When optimizing the unknown trainable parameters of a PMM in the form described above, the search process can be accelerated by applying unitary transformations to the matrices and eigenvectors. For each column or row dimension $n$ used in a PMM, we let $U_n$ be an arbitrary $n \times n$ unitary matrix. We multiply each eigenvector of length $n$ by $U_n$ and multiply each $n \times m$ matrix by $U_n$ on the left and $U_m^\dagger$ on the right. The unitary transformation leaves all outputs invariant while producing significant changes to the parameter values. We have found that combining random unitary transformations with local updating methods is very useful for accelerating parameter optimization in PMMs.

In order to provide an intuitive picture of the implicit functions being crafted by PMMs, we describe the connection between PMMs and a reduced basis method called eigenvector continuation (EC)[21–30]. Let us consider a family of Hermitian matrices $H(\{c_l\})$ whose elements are analytic functions of real input features, $\{c_l\}$. The EC calculation starts by picking some set of eigenvectors at selected values for the input features, and projects all vectors to the subspace spanned by these eigenvector "snapshots". The problem then reduces to finding the eigenvalues and eigenvectors of a much smaller Hermitian matrix $M(\{c_l\})$, whose elements are also analytic functions of $\{c_l\}$. The eigenvalues are roots of the characteristic polynomial, $\det[\lambda\mathbb{I} - M(\{c_l\})]$. If we approximate the dependence on the input features over some compact domain using polynomials, then each eigenvalue, $\lambda$, corresponds to the roots of a polynomial in $\lambda$ with coefficients that are themselves polynomials with respect to the input features. For all real values of the input features, $\lambda$ is analytic and bounded by the norm of $M(\{c_l\})$. Using only general arguments of analyticity, in ref. 30 it is shown that for $p$ input features and $N$ eigenvector snapshots, the error of the EC approximation diminishes as a decaying exponential function of $N^{1/p}$ in the limit of large $N$. For the case of eigenvalue problems, PMMs capture the essential features of EC calculations by proposing some unknown Hermitian matrix $M(\{c_l\})$ and then learning the matrix elements from data. In Methods, we discuss the connection further and consider examples where PMMs outperform EC.

Parametric matrix models can be designed to incorporate as much mathematical and scientific insight as possible or used more broadly as an efficient universal function approximator. It is useful to consider a class of PMMs where the matrix elements of the primary and secondary matrices are polynomial functions of the input features, $\{c_l\}$, up to some maximum degree $D$. In Methods, we prove a universal approximation theorem for such PMMs using only affine functions of the input features, corresponding to $D = 1$. In the numerical examples presented here, we show that these $D = 1$ PMMs achieve excellent

performance comparable to or exceeding that of other machine learning algorithms—while requiring fewer parameters. In addition, PMMs typically need fewer hyperparameters to tune as compared with other machine learning approaches, therefore requiring less fine-tuning and model crafting. This efficiency of description does not imply a loss of generality or flexibility. It is instead the result of a natural inductive prior that prioritizes functions with the simplest possible analytic properties. As the physics of eigenstates, dynamics, and transition probabilities of Hermitian systems has been well-studied in the field of quantum mechanics, the generality of such a PMM does not preclude its interpretability.

While matrix-based approaches have also been widely used in machine learning for dimensionality reduction[31–33], PMMs represent an approach based on physics principles and implicit functions derived from matrix equations. We first demonstrate the superior performance of PMMs for three scientific computing examples: multivariable regression, quantum computing, and quantum many-body systems. We then show the broad versatility and efficiency of PMMs on several supervised image classification benchmarks as well as hybrid machine learning when paired together with both trainable and pre-trained convolutional neural networks.

In this work we consider the number of trainable parameters as a concise and interpretable indicator of efficiency. This is not because the limiting factor in machine learning is the storage of model parameters. Instead it is because—for the forms of PMMs in this work and the ubiquitous feedforward neural network—training and inference complexity scales proportionally to the number of trainable parameters. This relation also holds for individual layers of significantly more advanced neural network architectures such as the self-attention layer which composes the transformer deep learning architecture, and which in turn plays a central role in large language models[34]. This is discussed further in Methods. Additionally in the context of emulating physical systems, where the trainable parameters may have physical significance and therefore the determination of the parameters is potentially as important as finding a model that reproduces the data, the number of trainable parameters provides a lower bound on the number of informative training examples necessary to identify sufficiently narrow neighborhoods of parameter solutions[35]. A direct comparison in the number of CPU hours or other empirical measures of computational cost between PMMs and existing techniques is not included in this work as such a comparison would be heavily biased in favor of methods with the benefit of decades of software and hardware optimization. We therefore present the number of trainable parameters as a useful metric indicative of both the computational cost and model size that will serve as a useful baseline for future work solidifying such comparisons.

## Results
### Regression
For our first benchmark, we have compared the performance of the general, basic, affine PMM for multivariable regression against several standard techniques, each of which were hyperparameter-tuned using grid search: Kernel Ridge Regression (KRR), Multilayer Perceptron (MLP), $k$-Nearest Neighbors (KNN), Extreme Gradient Boosting (XGB), Support Vector Regression (SVR), and Random Forest Regression (RFR). See, for example, ref. 36 for a description of most of these methods and ref. 37 for XGB. Full details on the grid searches are provided in Supplementary Tables 1–4. We consider two-dimensional test functions that include thirteen individual functions and two classes of functions that are described in Methods[38–42]. For each of these functions, we train a PMM with one primary Hermitian matrix, either $7 \times 7$ or $9 \times 9$, and form outputs using the three eigenvectors associated with the largest magnitude eigenvalues and Hermitian secondary matrices. We also test the performance on the NASA Airfoil dataset[43] of measurements of 2D and 3D airfoils in a wind tunnel and

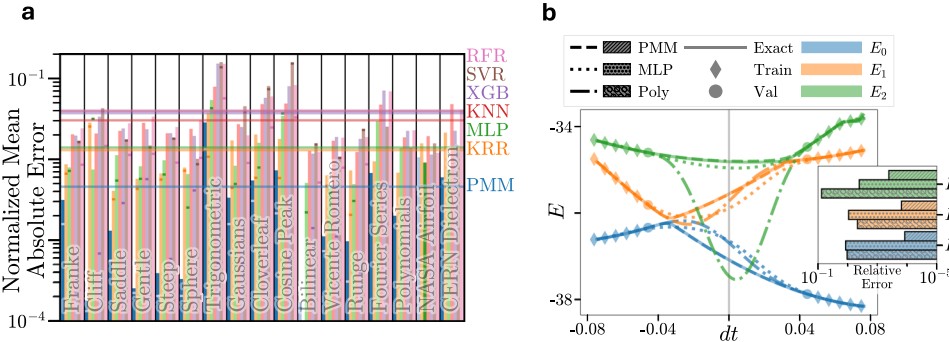

**Fig. 1 | PMM results for regression and Trotter extrapolation. a** Performance on regression problems. Normalized mean absolute error on withheld test data for the PMM (blue) compared against several standard techniques: Kernel Ridge Regression (KRR, orange), Multilayer Perceptron (MLP, green), $k$-Nearest Neighbors (KNN, red), Extreme Gradient Boosting (XGB, purple), Support Vector Regression (SVR, brown), and Random Forest Regression (RFR, pink). Normalized mean absolute error on provided training and validation data is shown for selected datasets as contrasting squares. Mean performance on withheld test data across all problems

are shown as horizontal lines. **b** Extrapolated Trotter approximation for quantum computing simulations. We plot the lowest three energies of the effective Hamiltonian for the one-dimensional Heisenberg model with DM interactions versus time step $dt$. We compare results obtained using a PMM (dashed), Multilayer Perceptron (MLP, dotted), and polynomial interpolation (Poly, dash-dotted). All training (diamonds) and validation (circles) samples are located away from $dt = 0$, where data acquisition on a quantum computer would be practical. The inset shows the relative error in the predicted energies at $dt = 0$ for the three models.

the CERN dielectron dataset[44] of 100,000 collision events. Similar to the approach used for the test functions, three eigenvectors of a $7 \times 7$ ($15 \times 15$) primary Hermitian matrix were used to form the output of the PMM for the NASA (CERN) dataset. We emphasize that other than the size of matrices and number of eigenvectors, the same PMM form was used in all regression tests here. Moreover, this form was not crafted using information about any data from any problem, but instead was designed as a general function approximator only. Full details, including the explicit form of the PMM used, are given in Methods.

In Panel a of Fig. 1, we show the normalized mean absolute error on withheld test data for the seventeen different regression examples. The normalized mean absolute error is the average absolute error divided by the largest absolute value in the unseen test data. The mean performance for all benchmarks is indicated by horizontal lines, and we see that the performance for PMMs is significantly better than that of the other methods, obtaining the lowest error for fifteen of the seventeen benchmarks. The functional form for KRR is ideally suited for bilinear functions, and its corresponding error for Bilinear lies below the minimum of the plot range in Panel a of Fig. 1. MLP has a slightly smaller error for the NASA Airfoil dataset. However, PMMs use at most an order of magnitude fewer trainable real parameters (`floats`) than the corresponding MLP for each benchmark example.

## Zero-error Trotter step extrapolation

For our second benchmark, we turn to a fundamental problem in quantum computing regarding the time evolution of a quantum system. The Hamiltonian operator $H = \sum_l H_l$ determines the energy of a quantum system, and the time evolution operator $U(dt)$ for time step $dt$ can be approximated as a product of terms $\prod_l \exp(-iH_l dt)$ with some chosen ordering. This product is known as a Trotter approximation[45], and higher-order improvements to the Trotter approximation are given by the Trotter-Suzuki formulae[46–48]. It is convenient to define an effective Hamiltonian $H_{eff}$ such that its time evolution operator $\exp(-iH_{eff}dt)$ matches the Trotter approximation for $U(dt)$. Using the Trotter approximation, the quantum computer can then find the eigenvalues and eigenvectors of $H_{eff}$. In the limit $dt \to 0$, $H_{eff}$ is the same as $H$. However, the number of quantum gate operations scales inversely with the magnitude of $dt$, and so a major challenge for current and near-term quantum computing is extrapolating observables to $dt = 0$ from values of $dt$ that are not very small in magnitude. Current state-of-the-art approaches to this problem utilize polynomial fitting techniques[49–51].

We consider a quantum spin system of $N = 10$ spins corresponding to a one-dimensional Heisenberg model with an antisymmetric spin-exchange term known as the Dzyaloshinskii-Moriya (DM) interaction[52,53]. Such spin models have recently attracted interest in studies of quantum coherence and entanglement[54–56]. For the PMM treatment of time evolution using this Hamiltonian, we replicate the matrix product structure of the Trotter approximation using a product of $9 \times 9$ unitary matrices, $\prod_l \exp(-iM_l dt)$. We then find the eigenvalues and eigenvectors of the Hermitian matrix $M_{eff}$ such that $\exp(-iM_{eff}dt) = \prod_l \exp(-iM_l dt)$. In Panel b of Fig. 1, we show the lowest three energies of the effective Hamiltonian, $H_{eff}$, versus time step $dt$. We show results for the PMM in comparison with a Multilayer Perceptron (MLP) with three hidden layers with 20 nodes each using the hyperbolic tangent activation function and polynomial interpolation (Poly). The depth, width, activation function, regularization strength, and learning rate for the MLP were found via grid search hyperparameter tuning. Full details of the grid search can be found in Supplementary Table 5. The training and validation samples are located away from $dt = 0$, where calculations on a current or near-term quantum computer are practical. The relative errors at $dt = 0$ for the predicted energies are shown in the inset, and we see that the PMM is more accurate than the two other methods for each of the low-lying energy eigenvalues and more than an order of magnitude better for the ground state $E_0$.

## Emulation of a quantum phase transition

For our third benchmark, we consider a quantum Hamiltonian for spin-1/2 particles on $N$ lattice sites with tunable anharmonic long-range two-body interactions called the anharmonic Lipkin-Meshkov-Glick (ALMG) model[57,58]. Here we take the anharmonicity parameter, $\alpha$, to be $\alpha = -0.6$. There is a second-order ground-state quantum phase transition in the two-particle pairing parameter, $\xi$, at $\xi = 0.2$, above which the ground state becomes exactly degenerate in the large $N$ limit and the average particle density in the ground state, $\langle \hat{n} \rangle / N$, becomes nonzero. For the calculations presented here, we take $N = 1000$. The details of the Hamiltonian are described in refs. 57,58. All training and validation data was taken away from the region of the phase transition. We employ a $9 \times 9$ PMM with primary matrices that form an affine latent-space effective Hamiltonian of the same form as the true underlying model as well as a secondary matrix that serves as the latent-space effective observable. The PMM is trained by optimizing the mean squared error between the lowest two eigenvalues of the

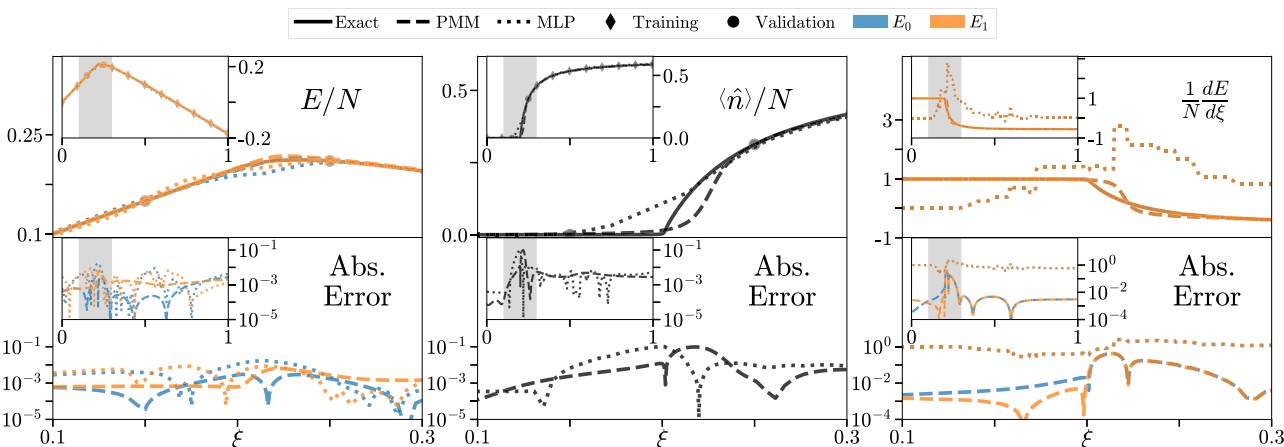

**Fig. 2 | ALMG model results.** Left Panel: Lowest two energy levels of the ALMG model versus $\xi$. We show PMM results compared with Multilayer Perceptron (MLP) results. The upper plots show the energies, and the lower plots show absolute error. The main plots show the region around the phase transition; the insets show the full domain where data was provided. Center Panel: Average particle density for the ground state of the ALMG model versus $\xi$. We show PMM results compared with Multilayer Perceptron (MLP) results. The upper plots show the average particle density, and the lower plots show absolute error. The main plots show the region around the phase transition; the insets show the full domain where data was provided. Right Panel: Derivative of the lowest two energy levels with respect to the control parameter $\xi$ as a function of $\xi$. We show PMM results compared with Multilayer Perceptron (MLP) results. The upper plots show the derivatives of the energies with respect to the control parameter, and the lower plots show absolute error. The main plots show the region around the phase transition; the insets show the full domain where data was provided. No data on the derivatives was provided to either model.

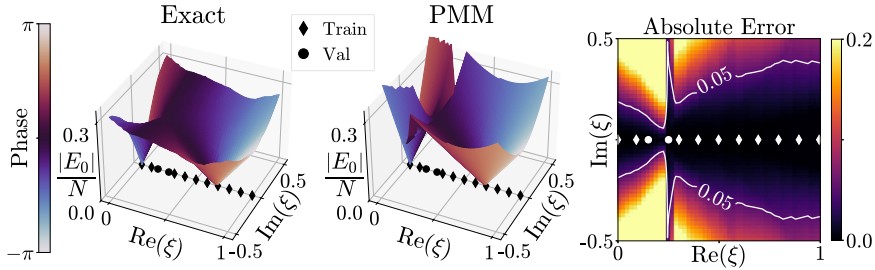

**Fig. 3 | Complex-valued ground state energy of the ALMG model for complex $\xi$.** We show PMM predictions for the complex-valued ground state energy for complex values of $\xi$, using training data at only real values of $\xi$. The left plot shows the exact results, the middle plot shows the PMM predictions, and the right plot shows the absolute error.

effective Hamiltonian and the expectation value of the effective observable in the ground state of the effective Hamiltonian to the data. A regularization term, proportional to the sum of differences between unity and the overlap between eigenvectors at successive values of $\xi$, is added to this loss function to encode for the possibility of a phase transition. Additionally, the secondary matrix in the PMM is constrained to be positive semi-definite, as this is true of the original observable. The same amount of physical information cannot be directly encoded in traditional neural network methods. The go-to and state-of-the-art neural network approaches for similar emulation problems utilize unmodified, standard architectures such as fully-connected feedforward neural networks or restricted Boltzmann machines[59–61]. As such, we compare our method to an MLP with a single input node, three hidden layers of 100, 100, and 10 neurons each with ReLU activations, and three output neurons which was trained by optimizing the mean squared error between the predictions and the data with a standard $l_2$ regularizer on the weights. The hyperparameters of the MLP including the weight of the regularizer, the activation function, widths of each layer, number of hidden layers, and learning rate were found via grid search hyperparameter tuning. Full details on the grid search are provided in Supplementary Table 6. In the left panel of Fig. 2, we show PMM results versus MLP results for the lowest two energy levels of the ALMG model versus $\xi$. In the center

panel of Fig. 2 we show PMM results versus MLP results for the average particle density in the ground state, $\langle \hat{n} \rangle / N$, versus $\xi$. In the right panel of Fig. 2 we show the derivative of the energy levels with respect to the control parameter $\xi$. Such derivatives are typically taken when searching for a phase transition in addition to, or in place of, an order parameter. We see that away from the phase transition, both models perform comparably. However, at the phase transition, only the PMM is able to accurately predict the sharp change in character of the system. This is especially prevalent in the derivatives of the energy levels.

In addition to superior numerical performance for scientific computing, PMMs also provide functionality for mathematical analysis that is difficult to match using other machine learning methods. The advantage for PMMs lies in its core design, which is based on mathematical equations that can be analytically continued into the complex plane without extra information. In Fig. 3, we show predictions using the same PMM as discussed above for the complex-valued ground state energy of the ALMG model for complex values of $\xi$. We emphasize that the PMM is trained using only real values of $\xi$. The left plot shows the exact results, the middle plot shows the PMM predictions, and the right plot shows the absolute error with contour lines outlining the region where the error is less than 0.05. The ability of the PMMs to accurately extrapolate the behavior of systems to complex parameters from purely real-valued−and therefore physical−data is invaluable for

**Table 1 | Supervised machine learning on image datasets**

| Dataset | Model | Acc. | float |
|---|---|---|---|
| MNIST Digits[86] | PMM | 97.38 | 4990 |
| | ConvPMM | 99.10 | 129416 |
| | DNN-2[87] | 96.5 | ~ 311650 |
| | DNN-3[87] | 97.0 | ~ 386718 |
| | DNN-5[87] | 97.2 | ~ 575050 |
| | GECCO[88] | 98.04 | ~ 19000 |
| | CTM-250[89] | 98.82 | 31750 |
| | CTM-8000[89] | 99.4 | 527250 |
| | Eff.-CapsNet[90] | 99.84 | 161824 |
| Fashion MNIST[91] | PMM | 88.58 | 16744 |
| | ConvPMM | 90.94 | 278280 |
| | GECCO[88] | 88.09 | ~ 19000 |
| | CTM-250[89] | 88.25 | 31750 |
| | CTM-8000[89] | 91.5 | 527250 |
| | MLP[92] | 91.63 | 2913290 |
| | VGG8B[92] | 95.47 | ~ 7300000 |
| | F-T DARTS[93] | 96.91 | ~ 3200000 |
| EMNIST Balanced[94] | PMM | 81.57 | 13792 |
| | ConvPMM | 85.95 | 349172 |
| | CNN[95] | 79.61 | 21840 |
| | CNN (S-FC)[95] | 82.77 | 13820 |
| | CNN (S-FC)[95] | 83.21 | 16050 |
| | HM2-BP[96] | 85.57 | 665647 |

We show PMM and ConvPMM results compared to other highly efficient methods on several image classification datasets. We present the accuracy on withheld test sets in percent and the number of trainable floating point parameters.

**Table 2 | Hybrid transfer learning results for ResNet50 combined with either a PMM or a feedforward neural network (FNN) head**

| Dataset | PMM | | FNN | |
|---|---|---|---|---|
| | float | Accuracy | float | Accuracy |
| CIFAR-10[97] | 95176 | $83.11^{+0.18}_{-0.27}$ | 271178 | $83.68^{+0.19}_{-0.45}$ |
| CIFAR-100[97] | 115570 | $56.17^{+0.23}_{-0.23}$ | 277028 | $55.05^{+0.66}_{-0.66}$ |
| SVHN[98] | 206920 | $65.28^{+0.20}_{-0.46}$ | 271178 | $65.22^{+0.35}_{-0.35}$ |
| STL-10[99] | 66880 | $86.14^{+0.40}_{-0.40}$ | 271178 | $85.94^{+0.17}_{-0.17}$ |
| Caltech256[100] | 227207 | $65.9^{+1.0}_{-1.0}1.4$ | 287233 | $75.48^{+0.30}_{-0.46}$ |
| CINIC-10[101] | 66880 | $72.93^{+0.11}_{-0.16}$ | 271178 | $72.91^{+0.23}_{-0.10}$ |

Each model was randomly initialized 10 times and trained for 10 epochs. The mean test accuracy over the trials are reported alongside the number of trainable floating point parameters. The pre-trained ResNet50 network uses ~ $23 \times 10^6$ trainable parameters. The positive (negative) uncertainty is the difference between the mean score and the mean of all scores greater (less) than the mean.

understanding and analyzing the mathematical structures giving rise to exceptional points, avoided level crossings, and phase transitions.

## Image classification

To demonstrate the efficacy of PMMs beyond scientific computing, for our fourth benchmark we address the problem of supervised image classification on several standard datasets. The field of computer vision has long been dominated by convolutional neural networks (CNNs), and still today, many recent machine learning architectures in computer vision make significant use of convolutions. However, it is often the case that without careful regularization, architecture refinements, or model crafting, these networks become massively over-parameterized. Here, we formulate a general PMM for the problem of image classification that is both accurate and highly efficient in the number of trainable parameters. The form of this PMM is not guided by any specific image data, but instead only by the fundamental treatment of images as data matrices and the desire to work with Hermitian matrices. While the form of the primary matrices differ from the PMMs used in the regression experiments, the form of the secondary matrices remains unchanged.

A natural encoding for images in the context of Hermitian matrix equations is to consider smaller fixed square or rectangular windows of each image, $W_l$. The row-wise and column-wise Gram matrices of the windows, $W_l W_l^\dagger$ and $W_l^\dagger W_l$ respectively, are combined using trainable transformations into a single primary matrix for each possible output class, as described in Methods. We then form bilinears of the eigenvectors of the primary matrices with trainable secondary matrices and use the resulting scalars to produce class probabilities for the input images. We train the model using complex-valued gradient descent to minimize the categorical cross-entropy loss function, a measure of the difference between the predicted and true probabilities. Full details of

the architecture are found in Methods. In Table 1, we show results for this pure PMM model versus the performance of other highly efficient methods on several longstanding image classification datasets. The datasets were chosen as they are well-established benchmarks still used in recent publications. We compare this introductory implementation of PMMs to neural-network-based approaches which have had nearly a decade to develop with the most recent of these datasets. We present the accuracy on withheld test sets in percent as well as the number of trainable floating point parameters. We see that in all cases, the performance of our method is comparable or superior to existing methods, despite using fewer parameters.

While we have thus far focused on standalone applications of PMMs, we expect there to be significant community interest in combining PMMs with other machine learning algorithms that have already demonstrated excellent performance and computational scaling. To demonstrate this application, we formulate a convolutional PMM (ConvPMM) by using the aforementioned image classification PMM to replace the head of several traditional, but complex-valued, convolutional neural network model architectures. All parameters in the ConvPMM, both the convolutional layers and the PMM head, can be trained simultaneously via backpropagation. The results for this model are also reported in Table 1 and demonstrate that traditional techniques can be combined with PMMs to achieve highly accurate and efficient models. Further details on the ConvPMM architecture are found in Methods.

To further demonstrate the capability of PMMs to integrate with existing neural networks, our fifth benchmark is a hybrid transfer learning model for image recognition, where a pre-trained convolutional neural network called ResNet50[62] is combined with a PMM. ResNet50 is used as a feature extractor and the PMM is used as a trainable classifier on these extracted features. The PMMs used were of the same form as those used for the regression experiments. The 2048 outputs of the spatially-pooled ResNet50 network form the input features and the softmax of the outputs form the class probability predictions. The number, rank, and size of matrices in the PMMs were chosen to accommodate the complexity, size, and number of classes in each dataset. Further details can be found in Methods. In Table 2, we show hybrid transfer learning results obtained for several different image datasets when combining ResNet50 with either a PMM or a feedforward neural network (FNN). We have performed 10 trials for each dataset with 10 training epochs each, and reported the mean test accuracy and number of trainable floating point parameters for each dataset and model. This experiment demonstrates that PMMs can be directly used with the encoded latent-space features from existing, pre-trained, purely-artificial-neural-network models.

## Discussion

We have presented parametric matrix models, a class of machine learning algorithms based on learning the underlying equations which govern data. PMMs have shown superior performance for several different scientific computing applications. The performance advantage over other machine learning methods is likely due to at least two major factors. The first is that PMMs can incorporate important mathematical structures such as operator commutation relations associated with the physical problem of interest. The second is that the forms of PMMs presented produce output functions with analytical properties determined by the eigenvalues and eigenvectors of the primary matrices. Using the properties of sharp avoided level crossings, these PMMs are able to reproduce abrupt changes in data without also producing the unwanted oscillations typically generated by other approaches.

While there are numerous studies that combine classical machine learning with quantum computing algorithms[63–65], the general form PMMs considered in this work use the matrix algebra of quantum mechanics as their native language. This unusual design also sets PMMs apart from other physics-inspired and physics-informed machine learning approaches[1–3]. Rather than imposing auxiliary constraints on the output functions, important mathematical structures such as symmetries, conservation laws, and operator commutation relations come directly from the known or supposed underlying equations. Once gate fidelity and qubit coherence are of sufficient quality for the operations required, the general form PMM can be implemented using techniques similar to those used in a field of quantum computing algorithms called Hamiltonian learning[66–68]. Although the primary computational advantage of PMMs is in the area of scientific computation, we have also shown that PMMs are universal function approximators that can be applied to general machine learning problems such as image recognition and can be readily combined with other machine learning approaches. For general machine learning problems, we have found that PMMs are competitive with or exceed the performance of other machine learning approaches when comparing accuracy versus number of trainable parameters. While the performance of PMMs for very large models has not yet been investigated, the demonstrated efficiency and hybrid compatibility of PMMs show the value of PMMs as a tool for general machine learning.

## Methods

Where not otherwise explicitly stated, we will use additional notation to clearly distinguish trainable-, fixed-, and hyper-parameters. Trainable parameters will be denoted by an underline, $\underline{x}$. Hyperparameters will be denoted by an overline, $\overline{\eta}$. And parameters fixed by the problem or data are written plainly, $s$. Optimization, or training, of the trainable parameters of a PMM can be accomplished by a global optimizer if the PMM is small enough or, in general, a modified version of the Adam gradient descent method[69] for complex parameters—as we have chosen in this work.

### Properties of PMMs

As described in the main text, the basic PMM form consists of primary matrices, $P_j$, and secondary matrices, $S_k$. All of the matrix elements of the primary and secondary matrices are analytic functions of the set of input features, $\{c_l\}$. The primary matrices are square Hermitian or unitary matrices, and their normalized eigenvectors, $\mathbf{v}_j^{(i)}$, are used to form bilinears with secondary matrices of the appropriate row and column dimensions to form scalar outputs of the form $\mathbf{v}_j^{(i)\dagger} S_k \mathbf{v}_j^{(i')}$. Making an analogy with quantum mechanics, we can view the set of eigenvectors used in this form of PMM as describing the possible quantum states of a system. The primary matrices $P_j$ are quantum operators simultaneously being measured, resulting in the collapse onto eigenvectors of the measured operators. The secondary matrices correspond to observables that measure specific properties of the quantum state or transitions between different quantum states. If we

ignore the irrelevant overall phase of each eigenvector, the total number of real and imaginary vector components that comprise the eigenvectors used in the PMM should be viewed as the number of dimensions of the latent feature space. This is analogous to the minimum number of nodes contained within one layer of a neural network.

The output functions of neural networks do not correspond to the solutions of any known underlying equations. Rather, they are nested function evaluations of linear combinations of activation functions. Many commonly-used activation functions have different functional forms for $x < 0$ and $x \geq 0$ and therefore are not amenable to analytic continuation. Most of the other commonly-used activation functions involve logarithmic or rational functions of exponentials, resulting in a more complicated analytic structure.

Let us consider a PMM that uses Hermitian primary matrices with matrix elements that are polynomials of the input features up to degree $D$. Suppose now that we vary one input feature, $c_l$, while keeping fixed all other input features. The output functions of $c_l$ can be continued into the complex plane, with only a finite number of exceptional points where the functions are not analytic. These exceptional points are branch points where two or more eigenvectors coincide. A necessary condition for such an exceptional point is that the characteristic polynomial of one of the primary matrices, $\det[\lambda \mathbb{I} - P_j(\{c_l\})]$, has a repeated root. This in turn corresponds to the discriminant of the characteristic polynomial equaling zero. If the primary matrix has $n \times n$ entries, the discriminant is a polynomial function of $c_l$ with degree $n(n-1)D$. We therefore have a count of $n(n-1)D$ branch points as a complex function of $c_l$ for each primary matrix. If a branch point comes close to the real axis, then we have a sharp avoided level crossing and the character of the output function changes abruptly near the branch point. Our count of $n(n-1)D$ branch points for each primary matrix gives a characterization of the Riemann surface complexity for the functions that can be expressed using a PMM.

For the case where the primary matrix is a unitary matrix, we can restrict our analysis to unitary matrices close to the identity and write $U = \exp(-iM)$, where $M$ is a Hermitian matrix. If we have a multiplicative parameterization for $U$ of the form $U = \prod_l U_l$, then we can write $\exp(-iM) = \prod_l \exp(-iM_l)$ and use the Baker-Campbell-Hausdorff expansion to relate $M$ to the products and commutators of $M_l$. This analysis is analogous to parameterizing the algebra of a Lie group.

A detailed characterization of analytic structure is not possible for neural network output functions. However, we can make the qualitative statement that neural network output functions reflect a change in slope, in some cases quite abruptly, as each of the activation function used in the network crosses $x = 0$. We can therefore compare our estimate of $n(n-1)D$ branch points for each primary matrix with the number of network nodes with activation functions.

**Computational and parameter scaling.** An instructive comparison between artificial neural networks and the PMM forms considered in this work can be made in the computational complexity of a single inference calculation as well as in the scaling of the number of trainable parameters in relation to the "expressivity" of each model. We quantify expressivity by the number of possible non-analytic points in the complex-valued output space of the model. Consider the case where there are $p$ input features and $q$ output values. A simplified MLP with $l$ hidden layers each composed of $m$ neurons requires $\mathcal{O}(lm^2)$ floating point multiplication operations to perform a single inference. Similarly, the same MLP is described by $\mathcal{O}(lm^2)$ trainable floating point values.

We compare the scaling of this simplified MLP with the two constructed model PMMs considered in this work, the affine eigenvalue PMM (AE-PMM) detailed in Methods section "Eigenvalue and Eigenstate Observable Emulation" and the more general affine observable PMM (AO-PMM) detailed in Methods section "Regression and

**Table 3 | Leading-order scaling for various properties of PMMs and a fixed-width MLP**

| Quantity | In terms of... | MLP | AE-PMM | AO-PMM |
|---|---|---|---|---|
| Non-analytic points, $\xi$ | Architecture hyperparameters | $lm$ | $n^2$ | $n^2$ |
| | Inference complexity $\chi$ | $\chi/m$ | $\chi/q$ | $\chi/q$ |
| | Trainable parameters $\Sigma$ | $\Sigma/m$ | $\Sigma/p$ | $\Sigma/(p+qr^2)$ |
| Inference complexity, $\chi$ | Architecture hyperparameters | $lm^2$ | $qn^2$ | $qr^2n^2$ |
| | Non-analytic points $\xi$ | $m\xi$ | $q\xi$ | $qr^2\xi$ |
| | Trainable parameters $\Sigma$ | $\Sigma$ | $\Sigma$ | $\Sigma$ |
| Trainable parameters, $\Sigma$ | Architecture hyperparameters | $lm^2$ | $pn^2$ | $(p+qr^2)n^2$ |
| | Non-analytic points $\xi$ | $m\xi$ | $p\xi$ | $(p+qr^2)\xi$ |
| | Inference complexity $\chi$ | $\chi$ | $\chi$ | $\chi$ |

The two constructed model PMMs considered in this work are shown: the affine eigenvalue PMM (AE-PMM, Methods section "Eigenvalue and Eigenstate Observable Emulation") and the affine observable PMM (AO-PMM, Methods section "Regression and Classification PMMs"). All models are considered to have $p$ input features and $q$ output values. Each of the $l$ hidden layers of the MLP has $m$ neurons. The size of the matrices in the PMMs is $n \times n$ and the number of eigenvectors used in the AO-PMM is denoted by $r$. We assume that $l \gg p$, $l \gg q$, and $p \sim q$.

Classification PMMs". In either form, let the primary matrix of the PMMs be $n \times n$. The computation of the full spectrum of the primary matrix in both forms requires $\mathcal{O}(n^3)$ multiplication operations in practice. However, if only a small subset of the spectrum, $r$ levels, is required then this complexity reduces to $\mathcal{O}(rn^2)$. In the case that the eigenvalues are the outputs, then $r = q$. In the case of the AO-PMM, in which each output is formed from taking bilinears of these $r$ eigenvectors with $\sim qr^2$ secondary matrices, the number of multiplication operations is $\mathcal{O}(qr^2n^2)$. The affine eigenvalue PMM needs only the $\mathcal{O}(pn^2)$ trainable values in the primary matrix. In contrast, the scaling of the number of trainable parameters in the AO-PMM is jointly dominated by the number of elements in the trainable secondary matrices; so the number of trainable parameters is $\mathcal{O}((p+qr^2)n^2)$.

These scaling results are summarized in Table 3 and allow for further analogies to be made between standard size hyperparameters in neural networks and PMMs. We see that, in the context of expressivity, the dimension of the PMM matrices, $n$, plays the same role as both the width and depth of the MLP. However, in the context of both the inference complexity and the number of trainable parameters for both PMM formulations, the dimension of the matrices functions comparably to only the width of the MLP. In the case of the AE-PMM, the number of outputs and number of input features each affect the scaling of the inference complexity and model size respectively in the same way that the number of layers in the MLP does. We emphasize that these quantities are fixed by the problem and thus the scaling of all relevant quantities of the AE-PMM are determined by a single hyperparameter, $n$. This single-hyperparameter property can be both an advantage—encouraging model simplicity—or a disadvantage, for example when more control over the model is desired or the number of input features is large. This disadvantage motivates the AO-PMM, for which the quantity $p + qr^2$ acts similarly to the number of layers in the MLP in the scaling of the complexity and number of trainable parameters. While this introduces an additional hyperparameter, it is important to note that $r$ and $n$ are not independent as $r$ must be less than or equal to $n$. Furthermore, typical values of $m$, $l$, and analogous hyperparameters for other neural networks like CNNs are in the range of $\mathcal{O}(10^1)$ to $\mathcal{O}(10^4)$. Such a wide range of possible values even for modest network sizes necessitates expensive hyperparameter tuning or aggressive model crafting. The analogous hyperparameters in PMMs have a much more limited range by comparison with typical values of $n$ and $r$ in the ranges of 5–25 and 1–5 respectively. We note that the number of hyperparameters in a general feedforward neural network, instead of the simplified fixed-width MLP considered here, can be substantially larger as the size and connectivity of each layer may be important hyperparameters. Moreover, all neural network

approaches rely on the choice of the activation function for each layer —for which there are, in principle, unlimited possibilities.

The results in Table 3 reproduce the known result that deep neural networks—those with many more layers than nodes per layer, $l \gg m$— are preferable to the alternative. This can be seen by the result that both the complexity and expressivity of the MLP scale linearly with the depth $l$, compared to quadratically and linearly respectively with $m$. By comparison, the expressivity of the two general PMMs considered both scale linearly with the complexity, regardless of what size hyperparameters are changed. This property of expressivity scaling proportionally to complexity is how we define an "efficient" universal function approximator in this work. Both the two general PMMs considered and the simplified MLP have this property.

Additional control over the scaling of PMMs can be gained by specifying the rank of the trainable matrices. By using low-rank matrices—as we have done for several of the examples in this work—the model can be made significantly more efficient. This is analogous to sparse layers and sparsity-promoting regularization in neural networks.

We note that the computational complexity of back-propagation for the MLP and either of the two PMMs considered scales proportional to the inference complexity of each model. Thus, the cost of training the models is equivalent, to leading order and up to constant factors, to the number of trainable parameters.

Although we have not applied PMMs to sequential data in this work, further comparisons with more recent and complex sequence transduction models provide additional context for the number of trainable parameters and computational complexities presented above. We focus on the scaling for a single layer of various architectures: recurrent (R), convolutional (C), self-attention (SA), and restricted self-attention (RSA). For an input sequence of $h$ inputs each of dimension $p_{\text{in}}$ a single layer transforms this into an output sequence of equal length, $h$, where each output has dimension $q_{\text{out}}$. In the case of the convolutional layer an additional hyperparameter $k$ denotes the size of the kernel. If the entire model were just a single layer, the total input (output) size $hp_{\text{in}}$ ($hq_{\text{out}}$) is roughly analogous to the number of input (output) features, $p$ ($q$), discussed above. However, we focus on the case in which the layer is an internal, or hidden, layer in a model where it is a common choice to set $p_{\text{in}} = q_{\text{out}} \equiv d$. The forward-pass computational complexity of each of these layers to leading order in architecture hyperparameters is[34]

$$\chi_R = hd^2, \qquad \chi_C = khd^2,$$
$$\chi_{SA} = h^2d + hd^2, \qquad \chi_{RSA} = rhd + rd^2. \qquad (1)$$

We compare this with the scaling of the number of trainable parameters in each architecture,

$$\Sigma_R = d^2, \qquad \Sigma_C = kd^2, \qquad \Sigma_{SA} = d^2, \qquad \Sigma_{RSA} = d^2, \qquad (2)$$

and note that the scaling for the computational complexity for each of these layers is—to leading order in architecture hyperparameters and up to constant factors—again the same as the scaling for the number of trainable parameters.

### Eigenvalue and eigenstate observable emulation

The simplest PMM in the context of emulating Hamiltonian systems is that of the affine PMM used for emulating energies and eigenstate properties. By affine, we mean that the dependence of the matrix elements on the input features is at most linear. Suppose the true underlying system is described by a Hamiltonian which is a function of $p$ features, $\{c_l : 1 \le l \le p\}$. Given data for some subset of the energy levels at some values of these parameters, we can emulate the energies with the following affine PMM,

$$M(\{c_l\}) = \underline{M_0} + \sum_{l=1}^{p} c_l \underline{M_l}, \qquad (3)$$

where $\underline{M_l}$ are $p + 1$ independent $\overline{n} \times \overline{n}$ Hermitian matrices. When the PMM has this simple form and the output function is a set of eigenvalues of $M(\{c_l\})$, we will refer to the PMM as an affine eigenvalue PMM (AE-PMM). An eigenvalue as a PMM output can be viewed as the expectation value of a secondary matrix that is the same as the primary matrix that produced the eigenvector. The hyperparameter $\overline{n}$ must be chosen to be large enough to accommodate all of the levels provided in the data as well as the degrees of freedom required by the data. The elements of each $\underline{M_l}$ are found by optimizing the mean squared error between the data and the eigenvalues of $M$ evaluated at the corresponding values of $\{c_l\}$. A suitable mapping between the energy levels of the PMM and the true levels must be used. Typically, the provided data contains some number of the lowest lying energies of the true system, in which case the mapping can be the identity. That is, the ground state of the PMM is compared with the ground state data, the first excited state of the PMM is compared with the first excited state data, and so on. More complex mappings may be used in the case that the data contain arbitrary energy levels.

This PMM can be trivially extended to emulate observables measured in the eigenstates of the original Hamiltonian as a function of the same $\{c_l\}$. Given data for some number of observables measured in some subset of the eigenstates of the original Hamiltonian, the PMM can accommodate this new information via the introduction of a secondary matrix for each observable in the data, $\underline{O_k}$. The loss function is modified to include the mean squared error between the data and expectation values of these secondary matrices in corresponding eigenstates of the PMM primary matrix. Weighting hyperparameters which control the strength of the energy error and observable error terms in the loss function may be included to further control the learning of the PMM. Constraints on these observables—such as ensuring a given observable commutes with the Hamiltonian, is positive semi-definite, or is a raising operator—can be incorporated directly by specific construction of the corresponding secondary matrices in the PMM. For example, to ensure that the secondary matrix that emulates a density observable is positive semi-definite it can be constructed as $\underline{O} = \underline{Q}^\dagger \underline{Q}$ where $\underline{Q}$ is the Hermitian matrix whose elements are fit to the data.

As mentioned in the main text, a simple and physically motivated regularizer arises naturally in this formulation. In most physical systems, one expects the eigenstates to change smoothly as a function of the Hamiltonian input features. Equivalently, one expects the overlap between an eigenstate at some level for some values of the

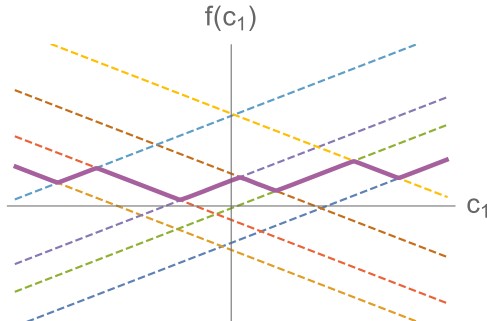

**Fig. 4 | Concatenated line segments for one input feature.** The thick line shows a particular eigenvalue $\lambda(c_1)$ that traces out a function composed of several concatenated line segments. The dashed lines show the affine functions $f_j(c_1) = a_j c_1 + b_j$ that describe the line segments.

Hamiltonian input features with the eigenstate for the same level at nearby values of the Hamiltonian input features to have magnitude near unity. We can encourage this behavior by adding a penalty term to the loss function which is proportional to the difference between neighboring eigenstate overlaps and unity. Let $\mathbf{v}$ be the vector whose entries are $1 - |\langle \psi_i(\{c_l\}) | \psi_i(\{c_l + \delta c_l\}) \rangle|^2$, where $|\psi_i\rangle$ is the $i^{th}$ eigenstate of the PMM, for all levels and feature space areas of interest. Then the penalty $\overline{\gamma} \|\mathbf{v}\|_{\overline{\alpha}}^{\overline{\alpha}}$ can be added to the previous loss function to encourage the smoothness of the eigenstates. Here, $\|\cdot\|_{\overline{\alpha}}^{\overline{\alpha}}$ is the $L^{\overline{\alpha}}$-norm to the power of $\overline{\alpha}$, where $\overline{\alpha}$ can be chosen to elicit the desired behavior. Most commonly, $\overline{\alpha} = 1$ encourages a few number of locations where the eigenstates are not smooth and $\overline{\alpha} = 2$ encourages a high average smoothness. The hyperparameter $\overline{\gamma}$ controls the strength of this regularization. Finally, it may be beneficial to modify the elements of $\mathbf{v}$ by normalizing them by each $\{\delta c_l\}$, as we have chosen to do in our implementation.

### Universal approximation theorem for PMMs

In Methods section "Eigenvalue and Eigenstate Observable Emulation", we discussed an affine eigenvalue PMM composed of one primary matrix that is a Hermitian matrix whose matrix elements are at most linear functions of the input features, $\{c_l\}$, and whose output corresponds to one particular eigenvalue of the primary matrix. In this section, we prove that even this most basic PMM is a universal function approximator. In more precise terms, we prove that for any compact domain of the input features, any uniform error tolerance $\epsilon > 0$, and any continuous real-valued function $f(\{c_l\})$, we can find an affine eigenvalue PMM that reproduces $f(\{c_l\})$ over the compact domain with uniform error less than $\epsilon$.

We start by proving an addition theorem for affine eigenvalue PMMs. Suppose that we have two AE-PMMs with output functions $f(\{c_l\})$ and $g(\{c_l\})$ corresponding to eigenvalues of primary matrices $P_f(\{c_l\})$ and $P_g(\{c_l\})$ respectively. We can define another affine eigenvalue PMM with output function $f(\{c_l\}) + g(\{c_l\})$ by constructing the tensor product of the two vector spaces and defining the new primary matrix as $P_{f+g}(\{c_l\}) = P_f(\{c_l\}) \otimes \mathbb{I} + \mathbb{I} \otimes P_g(\{c_l\})$.

Next, we prove the universal approximation theorem for affine eigenvalue PMMs with one input feature, $c_1$. Any continuous function $f(c_1)$ over a compact domain can be uniformly approximated to arbitrary accuracy using a concatenation of line segments with finite slope as shown by the thick line in Fig. 4. Let us label the line segments as $s_1$, $s_2$, …, $s_M$, where our ordering corresponds to ascending values for $c_1$. For each line segment, $s_j$, let us write the affine function that passes through $s_j$ as $f_j(c_1) = a_j c_1 + b_j$. We now construct a Hermitian matrix with the $j^{th}$ diagonal element given by $f_j(c_1)$. If the off-diagonal elements are all zero, then the eigenvectors are decoupled from each other and the

eigenvalues are the affine functions $f_j(c_1)$. We now make each of the off-diagonal matrix elements an infinitesimal but nonzero constant. The details are not important except that they mix the eigenvectors by an infinitesimal amount and produce sharp avoided level crossings. We now assume that each $|a_j|$ is large enough so that $f_j(c_1) = a_j c_1 + b_j$ does not intersect any line segment other than $s_j$. Let $n_b$ be the total number of affine functions $f_j(c_1)$ that pass below the first segment $s_1$. We note that for any line segment $s_{j'}$, exactly $n_b$ affine functions $f_j(c_1)$ pass below $s_{j'}$. We conclude that the $(n_b + 1)^{st}$ eigenvalue from the bottom of the spectrum will pass through all of the line segments $s_1, s_2, \ldots, s_M$. This completes the proof of the universal function approximation theorem for affine eigenvalue PMMs with a single input feature.

We now turn to the case with $p$ input features $c_1, \ldots, c_p$ with $p > 1$. Let us consider any monomial $c_1^{k_1} c_2^{k_2} \cdots c_p^{k_p}$ with total degree $k = k_1 + \cdots + k_p$. We can write the monomial as a finite linear combination of the form

$$c_1^{k_1} c_2^{k_2} \cdots c_p^{k_p} = \sum_{j=1}^{J} b_j \left[ \sum_{l=1}^{p} a_{j,l} c_l \right]^k, \quad (4)$$

where $J$ is a finite positive integer and all of the coefficients $b_j$ and $a_{j,l}$ are real numbers. This follows from the fact that

$$\left[ \sum_{l=1}^{p} a_{j,l} c_l \right]^k = \sum_{k_1 + k_2 + \cdots + k_p = k} \binom{k}{k_1, k_2, \ldots, k_p} (a_{j,1}^{k_1} a_{j,2}^{k_2} \cdots a_{j,p}^{k_p}) (c_1^{k_1} c_2^{k_2} \cdots c_p^{k_p}), \quad (5)$$

and the functions

$$\binom{k}{k_1, k_2, \ldots, k_p} (a_{j,1}^{k_1} a_{j,2}^{k_2} \cdots a_{j,p}^{k_p}) \quad (6)$$

are linearly independent functions of $\{a_{j,1}, a_{j,2}, \ldots, a_{j,p}\}$ for each distinct set of nonnegative integers $\{k_1, k_2, \ldots, k_p\}$. For sufficiently large but finite $J$, we can therefore write any homogeneous polynomial of total degree $k$ as a linear combination of $J$ polynomials of the form

$$\left\{ \left[ \sum_{l=1}^{p} a_{1,l} c_l \right]^k, \left[ \sum_{l=1}^{p} a_{2,l} c_l \right]^k, \ldots, \left[ \sum_{l=1}^{p} a_{J,l} c_l \right]^k \right\}, \quad (7)$$

where each of the coefficient values $\{a_{j,l}\}$ are fixed. Further literature on this topic and the minimum integer $J$ required can be found in ref. [70–72].

Each term in the outer sum of Eq. (4) has the form

$$b_j \left[ \sum_{l=1}^{p} a_{j,l} c_l \right]^k. \quad (8)$$

For each index $j$, this is a function of a single linear combination of input features. We can therefore construct an AE-PMM for each $j$ to uniformly approximate Eq. (8) to arbitrary accuracy. By the addition theorem for AE-PMMs, we can perform the sum over $j$ and uniformly approximate any monomial $c_1^{k_1} c_2^{k_2} \cdots c_p^{k_p}$ and therefore any polynomial. By the Stone-Weierstrass theorem[73,74], we can uniformly approximate any continuous function of the input features $\{c_l\}$ over any compact domain using polynomials. We have therefore proven that any continuous function of the input features $\{c_l\}$ over any compact domain can be uniformly approximated to arbitrary accuracy by AE-PMMs.

We define a unitary affine eigenvalue PMM to be a generalization of the affine eigenvalue PMM where the output is an eigenvalue of single primary matrix that is the exponential of the imaginary unit times a Hermitian matrix composed of affine functions of the input features. The proof of the universal approximation theorem for unitary affine eigenvalue PMMs is analogous to the proof for affine eigenvalue PMMs. The precise statement is that for any compact domain of the input features, any uniform error tolerance $\epsilon > 0$, and any continuous real-valued function $f(\{c_l\})$, we can find a unitary affine eigenvalue PMM that reproduces $\exp[if(\{c_l\})]$ over the compact domain with uniform error less than $\epsilon$.

Analogous to the addition theorem for affine eigenvalue PMMs, we can prove the multiplication theorem for unitary affine eigenvalue PMMs. Suppose that we have two unitary affine eigenvalue PMMs with output functions $f(\{c_l\})$ and $g(\{c_l\})$ corresponding to primary matrices $P_f(\{c_l\})$ and $P_g(\{c_l\})$ respectively. We can define another unitary affine eigenvalue PMM with output function $f(\{c_l\})g(\{c_l\})$ by constructing the tensor product of the two vector spaces and defining the new primary matrix as $P_{f \cdot g}(\{c_l\}) = P_f(\{c_l\}) \otimes P_g(\{c_l\})$.

For unitary affine eigenvalue PMMs with one input feature $c_1$, the proof of the universal approximation theorem uses exponentials of affine functions multiplied by the imaginary unit. We assign $f_j(c_1) = \exp\left[i\left(a_j c_1 + b_j\right)\right]$ to the $j^{th}$ diagonal entry of the primary matrix. Analogous to the monomial $c_1^{k_1} c_2^{k_2} \cdots c_p^{k_p}$ for affine eigenvalue PMMs with more than one input feature, for unitary affine eigenvalue PMMs we approximate the exponentiated monomial $\exp\left(i c_1^{k_1} c_2^{k_2} \cdots c_p^{k_p}\right)$. The other steps in the proof are straightforward and analogous to the derivations for the affine eigenvalue PMMs. Having proven that affine eigenvalue PMMs and unitary affine eigenvalue PMMs are universal function approximators, we conclude that all PMMs of the basic forms described in the main text with Hermitian or unitary primary matrices are universal function approximators.

## Comparison of PMMs and eigenvector continuation

We consider the problem of finding the eigenvalues and eigenvectors for a family of Hermitian matrices, $H(\{c_l\})$, whose elements are analytic functions of real input features, $\{c_l\}$. The reduced basis approach of eigenvector continuation[21] (EC) is well suited for this problem. We select some set of eigenvector snapshots at training points $\left\{ \{c_l^{(1)}\}, \{c_l^{(2)}\}, \cdots \right\}$. After projecting to the subspace spanned by these snapshots, we solve for the eigenvalues and eigenvectors of the much smaller Hermitian matrix $M(\{c_l\})$, whose elements are also analytic functions of $\{c_l\}$. While this works very well for many problems, there is a general problem that traditional reduced basis methods such as EC can lose accuracy for systems with many degrees of freedom. In such cases, the number of snapshots may need to scale with the number of degrees of freedom in order to deliver the same accuracy.

In the main text, we have noted that PMMs can reproduce the performance of EC by learning the elements of the matrices $M(\{c_l\})$ directly from data. In the following, we present an example where PMMs perform better than EC for a problem with many degrees of freedom. We consider a simple system composed of $N$ non-interacting spin-1/2 particles with the one-parameter Hamiltonian

$$H(c) = \frac{1}{2N} \sum_{i}^{N} (\sigma_i^z + c \sigma_i^x). \quad (9)$$

Here, $\sigma_i^z$ and $\sigma_i^x$ are the $z$ and $x$ Pauli matrices for spin $i$, respectively. We see in Fig. 5, that the EC method has difficulties in reproducing the ground state energy $E_0$ for large $N$ using five training points, or snapshots, which here are the eigenpairs obtained from the solution of the full $N$-spin problem. The PMM representation does not have this problem and is able to exactly reproduce $E_0$ for all $N$ using a learned $2 \times 2$ matrix model only of the form $M(c) = (\sigma^z + c \sigma^x)/2$. While this particular example is a rather trivial non-interacting system, it illustrates the general principle that PMMs have more freedom than traditional

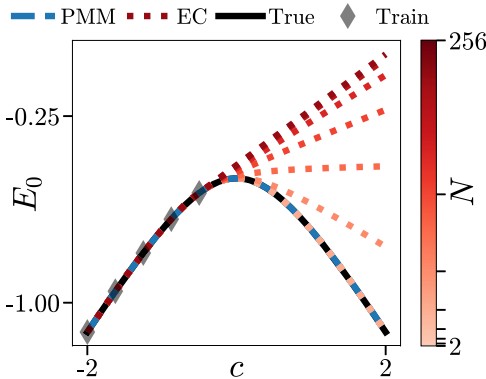

**Fig. 5 | Comparison of PMM and EC results for ground state energy extrapolation.** We show results for a $2 \times 2$ PMM (dashed blue) and EC (dotted red) with 5 training samples on the task of extrapolating the ground state energy of a system of $N$ non-interacting spins. The exact ground state energy is shown in solid black.

reduced basis methods. They do not need to correspond to a projection onto a fixed subspace, and this allows for efficient solutions to describe a wider class of problems.

It should be noted that for this simple example, the extrapolation problems for the EC calculation can be cured by computing a formal power series expansion around $c = 0$ and truncating the series up to any finite order in $c$. However, it is both notable and convenient that the problem never arises in the PMM framework and so no such additional analysis is needed.

### Regression and classification PMMs

We address the general problem of regression and classification in machine learning where the model learns to reproduce feature-label pairs $\{(\mathbf{x}_i, \mathbf{y}_i) : \mathbf{x}_i \in \mathbb{R}^p, \mathbf{y}_i \in \mathbb{R}^q\}$ and generalize to unseen data. Regression and classification are fundamentally the same problem, differentiated only by the nature of the labels $\{\mathbf{y}_i\}$ and therefore suitable choices of loss functions.

The simplest PMM for this task uses a primary matrix which is affine in the input features, as shown previously in Eq. (3). This is the form we have chosen for our regression experiments as well as our hybrid transfer learning experiments. The form of the primary matrix or matrices can be modified to accommodate known properties of the data, for example the image classification PMM discussed in Methods section "Image Classification PMM".

The first $\bar{r}$ eigenvectors associated with the largest magnitude eigenvalues form bilinears with Hermitian secondary matrices $\Delta_{kij} \in \mathbb{C}^{\bar{n} \times \bar{n}}$ where $1 \le k \le q$ and $1 \le i, j \le \bar{r}$ and $\Delta_{kij} = \Delta_{kji}$ to produce the output of the PMM. That is, for each output index $k$ and each pair of eigenvector indices $i, j$ there is an independent secondary matrix $\Delta_{kij}$. These bilinears—which can be thought of as expectation values (transition amplitudes) when $i = j$ ($i \ne j$)—are summed together with a trainable bias vector $\underline{\mathbf{g}} \in \mathbb{R}^q$ and a fixed bias proportional to the spectral norm of the secondary matrices to form the output of the PMM,

$$z_k = \underline{g_k} + \sum_{i,j=1}^{\bar{r}} |\mathbf{v}^{(i)\dagger} \underline{\Delta_{kij}} \mathbf{v}^{(j)}|^2 - \frac{1}{2} \left\| \underline{\Delta_{kij}} \right\|_2^2. \tag{10}$$

Equivalently, the sum may be restricted to $i \le j$ for efficiency. The fixed bias term is a deliberate addition to ensure that the output of the PMM is both unbounded and invariant under suitable unitary transformations of the trainable matrices. This output vector may be augmented by fixed or trainable activation functions, such as the softmax function in the case of classification.

The form of this PMM—which we call the affine observable PMM (AO-PMM) due to the analogy with observables and transition probabilities in quantum mechanics—is not motivated by any specific data, but instead purely by the desire to generalize the affine eigenvalue PMM described in Methods section "Eigenvalue and Eigenstate Observable Emulation" to cases with multiple outputs of arbitrary algebraic order.

**Regression experiments.** We have compared the performance of PMMs for multivariable function regression against several standard techniques: Kernel Ridge Regression (KRR), Multilayer Perceptron (MLP), $k$-Nearest Neighbors (KNN), Extreme Gradient Boosting (XGB), Support Vector Regression (SVR), and Random Forest Regression (RFR) (see for example ref. [36] for many of these methods and[37] for XGB). We have considered thirteen different two-dimensional test functions from refs. [38–42] as well as two classes of functions (Fourier series and polynomials) with exact forms given in Supplementary Table 7. For these functions the dataset consisted of a 200 point training set sampled from a uniform distribution and 10,000 point test set drawn from a grid with uniform spacing. For the classes of functions, 1000 functions were sampled and the mean performance for each model is reported. For each experiment, 10% of the training set was used as a validation set for the PMM, where the full training set was used for the other machine learning models optimized using $k$-fold cross-validation and grid search for hyperparameter tuning. For the two-dimensional test functions 10-fold cross validation was used, and the remaining experiments used 5-fold cross validation. Nearly all of the relevant hyperparameters for the non-PMM methods were tuned as follows: (KRR) the regularization strength, the kernel function, and the kernel function parameter if applicable; (MLP) the architecture by means of the number of layers and the number of nodes in each layer—not necessarily fixed-width—the activation function, the regularization strength, and the learning rate; (KNN) the number of neighbors, the weighting method, and the size of the leaves in the constructed tree; (XGB) the number of estimators, maximum depth, and learning rate; (SVR) the kernel function, the kernel parameter, and regularization strength; and (RFR) the number of estimators, maximum depth, and minimum number of samples required to split a node. Full details on the parameters searched are provided in Supplementary Tables 1–4. For the thirteen test functions (two classes of functions) a $7 \times 7$ ($9 \times 9$) primary matrix with $\bar{r} = 3$ PMM was tested against the other machine learning models. Finally, two standard regression datasets consisting of real-world data—the NASA airfoil[43] and CERN dielectron[44] datasets—were used to test the performance of PMMs. For these datasets, 35% of the data was used as the training set. For the NASA (CERN) dataset a $7 \times 7$ ($15 \times 15$) primary matrix with $\bar{r} = 3$ PMM was used.

**Image classification PMM.** To maximize efficiency in the task of image classification, we reformulate the primary matrices of the PMM to take advantage of the properties of images. A typical non-convolutional method will take the input features to be the vectorized, or flattened, images. Instead, we consider surrogates for the row-wise and column-wise correlation matrices of fixed windows of the images. This formulation yields a natural interpretation of images as the principle components of their constituent windows.

Given a grayscale image $X \in \mathbb{R}^{n \times m}$—or a color image whose color channels are compressed to two components using a method such as Independent Component Analysis and encoded as the real and imaginary parts of complex numbers, $X \in \mathbb{C}^{n \times m}$—we select $\overline{w}$ windows of shape $\overline{s_l} \times \overline{t_l}$, $l = 1, 2, \ldots, \overline{w}$ from the image. The area of the image that these windows cover can overlap or be entirely disjoint. For each window, we use the associated part of the image, $W_l \in \mathbb{C}^{\overline{s_l} \times \overline{t_l}}$, to calculate the row-wise and column-wise Gram matrices, $W_l W_l^\dagger$ and $W_l^\dagger W_l$ respectively, as efficient surrogates for the row-wise and column-wise correlation matrices. These matrices are uniformly normalized

element-wise such that the maximum magnitude of the entries is 1.0 before finally the diagonal elements are set to unity. Denote these post-normalized matrices by $R_l \tilde{\propto} W_l W_l^\dagger \in \mathbb{C}^{\overline{s_l} \times \overline{s_l}}$ and $C_l \tilde{\propto} W_l^\dagger W_l \in \mathbb{C}^{\overline{t_l} \times \overline{t_l}}$. These Hermitian matrices encode much of the information of the original image. This process of encoding the image as a set of $R_l$ and $C_l$ can be done either as a preprocessing step or included in the inference process of the PMM. This allows the PMM to be trained efficiently while still being able to operate on new previously-unseen data.

Using this Hermitian matrix encoding for images, we form the primary matrices of the image classification PMM in two steps. First, for each window we apply a quasi-congruence transformation using a trainable matrix $\underline{K_l} \in \mathbb{C}^{\overline{a} \times \overline{s_l}}$ ($\underline{L_l} \in \mathbb{C}^{\overline{a} \times \overline{t_l}}$) to $R_l$ ($C_l$) and sum over the windows to form a single Hermitian matrix, $M \in \mathbb{C}^{\overline{a} \times \overline{a}}$, which contains the latent-space features of the image,

$$M = \sum_l^{\overline{w}} \underline{K_l} R_l \underline{K_l}^\dagger + \underline{L_l} C_l \underline{L_l}^\dagger. \tag{11}$$

We describe the terms $K_l R_l K_l^\dagger$ and $L_l C_l L_l^\dagger$ as quasi-congruence transformations since $K_l$ and $L_l$ are not necessarily square matrices and thus these transformations are not exactly congruence transformations. With $M$ constructed, we apply another quasi-congruence transformation for each possible class output using trainable matrices, $\underline{D_k} \in \mathbb{C}^{\overline{b} \times \overline{a}}$, and add trainable Hermitian bias matrices, $\underline{B_k} \in \mathbb{C}^{\overline{b} \times \overline{b}}$, to form a primary matrix for each output,

$$H_k = \underline{D_k} M \underline{D_k}^\dagger + \underline{B_k}, \quad k = 1, 2, \ldots, q. \tag{12}$$

These primary matrices represent the latent-space class-specific features of the image. The eigensystems of these primary matrices are used to form predictions in a nearly identical way to the regression PMM shown in Eq. (10). However, for the image classification PMM described thus far the bilinears with the trainable secondary matrices need to account for the $q$ different primary matrices and so Eq. (10) becomes

$$z_k = \underline{g_k} + \sum_{i,j=1}^{\overline{r}} \left| \mathbf{v}_k^{(i)\dagger} \underline{\Delta_{kij}} \mathbf{v}_k^{(j)} \right|^2 - \frac{1}{2} \left\| \underline{\Delta_{kij}} \right\|_2^2, \quad k = 1, 2, \ldots, q, \tag{13}$$

where $\mathbf{v}_k^{(i)}$ is the $i^{\text{th}}$ eigenvector for primary matrix $H_k$. Finally, these outputs are converted to predicted class probabilities by means of a standard softmax with temperature $\overline{\tau}$,

$$\rho_k = \text{softmax}(z_k/\overline{\tau}) = \frac{\exp(z_k/\overline{\tau})}{\sum_{k'=1}^q \exp(z_{k'}/\overline{\tau})}. \tag{14}$$

This image classification PMM algorithm is summarized diagrammatically in Fig. 6 for the example of a $q = 2$ class dataset. We note that the form of the primary matrices in this PMM was not motivated by any specific data but rather by the natural interpretation of images as data matrices where the information is contained within a relatively small number of principle components.

**Convolutional image classification PMM.** We demonstrate the ability for existing neural network methods to be combined with PMMs, including the ability for gradients to be propagated through the PMM, by constructing and training a convolutional image classification PMM (ConvPMM). The ConvPMM is built on the image classification PMM described in Methods section "Image Classification PMM" with the normalization of the row- and column-wise Gram matrices skipped. The ConvPMM uses a trainable complex-valued convolutional neural network to compute filtered complex-valued "images" which the image classification PMM then processes. The architecture of the convolutional layers of the ConvPMM used for the

MNIST-Digits dataset consisted of four layers of 64, 32, 16, and 8 filters of size $3 \times 3$ with a stride of 1 and a ReLU activation function. The first two layers used "valid" padding while the last two layers used "same" padding. Panel a of Fig. 7 shows the convolutional layer architecture used in the ConvPMM for the MNIST-Digits dataset. For the Fashion-MNIST and EMNIST-Balanced datasets, the number of filters in each layer was doubled and an additional layer with 8 filters of size $3 \times 3$, a stride of 1, and "same" padding was added to the end.

**Hybrid transfer learning with ResNet50.** To demonstrate the ability for PMMs to be used in conjunction with and to complement established machine learning models, we have performed experiments with a hybrid transfer learning model built on ResNet50[62]. ResNet50 is a deep convolutional neural network that has been pre-trained on the ImageNet dataset of over one million images. For each of the datasets used in the experiments, the ResNet50 model—with a suitably-shaped input, $6 \times$ upsampling layer, and final classification layer replaced with a global spatial pooling layer as shown in Panel b of Fig. 7—was used to extract the features from the images. This resulted in a feature vector of length 2048 for each image. A feedforward neural network (FNN) with the architecture shown in Panel c of Fig. 7 as well as a PMM of the form described in Methods section "Regression and Classification PMMs", with a softmax on the outputs, were trained on these features. This can equivalently be thought of as a single self-contained model with several frozen, pre-trained traditional neural network layers followed either by more, trainable, traditional neural network layers (FNN) or by a trainable PMM acting as the final layer.

The sizes of the layers in the FNN were fixed and the dropout percentage hyperparameter was tuned via a grid search using the CIFAR-10 dataset. The sizes of the matrices and number of eigenvectors used in the PMM were chosen such that the number of trainable parameters was less than the corresponding FNN. For each dataset, each model was randomly initialized 10 times and trained for 10 epochs with the Adam optimizer[69] using the categorical cross-entropy loss function. For datasets that were not pre-split into training and validation sets, 10% of the training set was used as a validation set. The number of epochs was chosen such that training converged for all models. The top-1 accuracy of the models was calculated on the provided test sets and the mean with asymmetric uncertainties was reported. The positive (negative) uncertainty was calculated as the difference between the mean accuracy and the mean of all accuracies greater (less) than the mean accuracy.

**Zero-error Trotter step extrapolation**
There are several efficient algorithms that determine energy levels on a quantum computer using the complex phases produced during the time evolution of quantum states[75–83]. We consider the problem of extrapolating to $dt = 0$ given data of energies for $dt \leq \pi/\|H\|_2^2$. The Hamiltonian considered in this work is the one-dimensional Heisenberg model with an antisymmetric spin-exchange term known as the Dzyaloshinskii-Moriya (DM) interaction term with periodic boundary conditions.

$$\begin{aligned}
H = &\, B \sum_i^N r_i \sigma_i^z \\
&+ J \sum_i^N (\sigma_i^z \sigma_{i+1}^z + \sigma_i^x \sigma_{i+1}^x + \sigma_i^y \sigma_{i+1}^y) \\
&+ D \sum_i^N (\sigma_i^x \sigma_{i+1}^y - \sigma_i^y \sigma_{i+1}^x)
\end{aligned} \tag{15}$$

Where $r_i$ are random real numbers ranging from $[0, 1)$, and we have chosen $B = 2$, $J = 2$, and $D = 0.65$. The Trotter approximation for this underlying Hamiltonian consists of five terms: the external field term

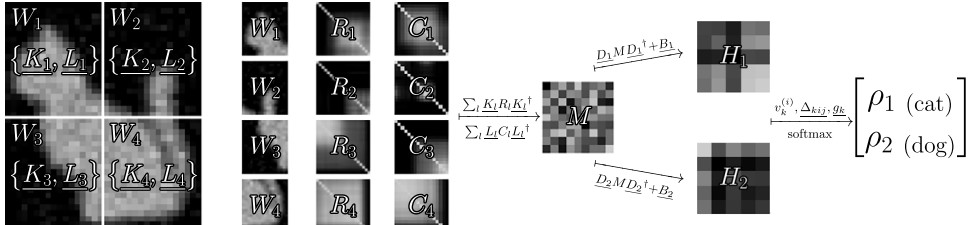

**Fig. 6 | Diagram of the image classification PMM algorithm.** We conceptually illustrate the inference process of the image PMM in the context of classifying images of dogs and cats. We start with the original image divided into four rectangular windows, $W_1, ..., W_4$, with trainable quasi-congruence transformation matrices $\{K_1, L_1\}, ..., \{K_4, L_4\}$. From each window the normalized row- and column-wise Gram matrices, $R_1, ..., R_4$ and $C_1, ..., C_4$, are calculated and summed to form the latent space feature encoding matrix $M$. Additional trainable quasi-congruence transformation matrices, $D_1$ and $D_2$, are applied and added to trainable Hermitian bias matrices, $B_1$ and $B_2$, to form the class-specific latent-space feature matrices, $H_1$ and $H_2$, which are the primary matrices of the PMM. Finally, the eigensystem of these primary matrices are used to form bilinears with the secondary matrices of the PMM before finally a softmax is applied to convert the predictions to probabilities.

**a**

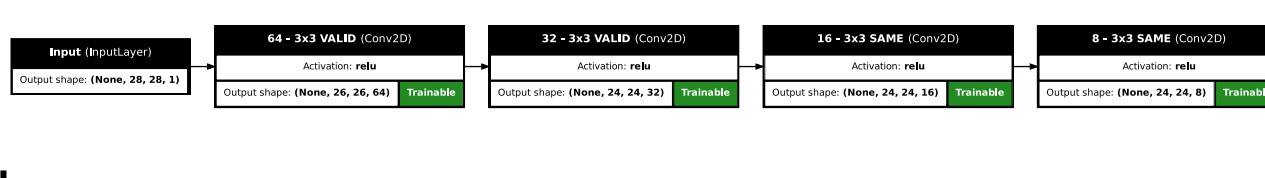

**b**

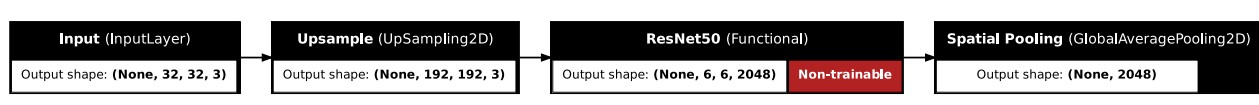

**c**

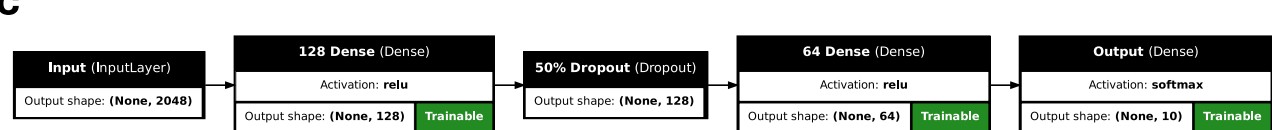

**Fig. 7 | Diagrams showing architectures. a** Diagram of the convolutional layer architecture used in the ConvPMM for the MNIST-Digits dataset. The architecture consists of four layers of 64, 32, 16, and 8 trainable complex-valued filters of size $3 \times 3$ with a stride of 1 and a ReLU activation function. The first two layers use "valid" padding while the last two layers use "same" padding. **b** Model diagram for the frozen, pre-trained ResNet50 model used as a feature extractor in the hybrid transfer learning experiments. The input shape is determined by the dataset, which is $32 \times 32 \times 3$ for the CIFAR-10 dataset used in this figure as an example. The output shape is 2048 for the extracted feature vector. **c** Model diagram for the trainable feedforward neural network (FNN) used in the hybrid transfer learning experiments. The output shape of the final layer is determined by the number of classes in the dataset, which is 10 in this figure as an example.

and each of the two interaction terms split by parity. We replicate this structure using the unitary eigenvalue PMM described in Methods section "Properties of PMMs",

$$U(dt) = \prod_l^5 \exp(-iM_l dt) \qquad (16)$$

where $M_l$ are the five independent $\bar{n} \times \bar{n}$ Hermitian matrices of the PMM which form the unitary primary matrix. The $q$ desired energies, $\{E_k\}$, are determined by setting $\exp(-iE_k dt)$ equal to the eigenvalues of $U(dt)$. As mentioned in Methods section "Eigenvalue and Eigenstate Observable Emulation", one must consider an appropriate mapping between the energy levels of the PMM and the true levels of the data. The elements of $M_l$ are found by optimizing the mean squared error between the data and the energies evaluated at the corresponding values of $dt$. Twelve points were generated as a training set for each energy level. The closest point to $dt = 0$ was used as a validation point for the PMM. The MLP was optimized through leave-one-out cross-

validation and hyperparameters were tuned via grid search. The polynomials were fit to all the available data.

## Data availability
All of the data produced in association with this work have been stored and are publicly available in this Code Ocean capsule.

## Code availability
All of the codes produced in association with this work have been stored and are publicly available in this Code Ocean capsule.

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

## Acknowledgements

We are grateful for discussions with Pablo Giuliani and Kyle Godbey, who have been developing methods that combine reduced basis methods with machine learning, and they have written several papers with collaborators since the posting of our original manuscript[84,85]. We also thank Scott Bogner, Wade Fisher, Heiko Hergert, Caleb Hicks, Felix Köhler, Yuan-Zhuo Ma, Jacob Watkins, and Xilin Zhang for useful discussions and suggestions. This research is supported in part by U.S. Department of Energy (DOE) Office of Science grants DE-SC0024586, DE-SC0023658 and DE-SC0021152. P.C. is supported by the Michigan State University (MSU) University Distinguished Fellowship (UDF) Program and was partially supported by the U.S. Department of Defense (DoD) through the National Defense Science and Engineering Graduate (NDSEG) Fellowship Program. M.H.-J. is partially supported by U.S. National Science Foundation (NSF) Grants PHY-1404159 and PHY-2310020. Da.L. is supported by the Institute of Information & Communications Technology Planning & Evaluation (IITP) grant funded by the Korean Government (MSIT) (No. RS-2024-00457882, National AI Research Lab Project). De.L. is partially supported by DOE grant DE-SC0013365, NSF grant PHY-2310620, and SciDAC-5 NUCLEI Collaboration.

## Author contributions

P.C. and D.J. carried out the theoretical and numerical analyses. De.L. contributed to the theoretical analyses. M.H.-J., Da.L., and De.L. supervised the work. All authors contributed to the discussion of the results and the manuscript.

## Competing interests

The authors declare no competing interests.
