## [Transparent Peer Review file · Nature Communications]

Parametric Matrix Models

Corresponding Author: Professor Dean Lee

Version 0:

Reviewer comments:

Reviewer #1

(Remarks to the Author)

I had reviewed this work in previous submission (Referee #1). I would like to thank the authors for this revised version of their manuscript, which addresses all of my comments.

More specifically, I appreciate the changes to the text emphasizing the kind of problems where PMMs shine, and the clarifications about the grounds of comparison with other approaches.

Regarding my second point, the computational cost of PMMs, I agree with the idea that a direct comparison of CPU hours would be unfair to PMMs, although I still believe that an indicative number (with the appropriate disclaimer) would be to the benefit of the reader, as the total lack of figures might seem indicative of the prohibitive cost of this approach. However, I do appreciate the expanded section S1.1, which at least gives an idea of the computational scaling.

In conclusion, I still believe this to be an interesting approach, and the revised manuscript to be worth publishing in this new venue in its current form.

(Remarks on code availability)

I had previously reviewed the code and still believe it to be sufficiently clear and reproducible.

Reviewer #2

(Remarks to the Author)

I thank the authors for their detailed and thorough response. I am satisfied with many of the responses and believe this paper is a good fit for Nature Communications. Please assume that any points I do not follow up on below are no longer points of concern.

I am happy with the aspect of the work that focuses on PMMs as specialized to situations where “known or supposed governing equations” are given trainable parameters. I want to follow up on some of the work using PMMs as a general method.

- I still struggle with the “efficient” universal function approximator statements. How is expressivity defined in the statement that expressivity grows linearly with complexity? And what does complexity mean in this statement? I have never heard of the concept of an “efficient” universal function approximator, and if the authors are claiming something that sounds fairly strong, it should be better justified.

- I am unclear whether the hyperparameter grid search was done by the authors or by previous works. If done by the authors, I would suggest providing more information about the grid search performed. The response currently states that grid search was done over “the weight of the regularizer, the activation functions, widths of each layer, and number of hidden layers”. How many choices for each was used? More conventionally, one would not sweep over choices of activation functions and instead include the learning rate. I don’t mean to dwell on this. The more important point is that when reporting performance of other methods where the numbers do not come from extant work, it’s important to give confidence a best effort was made.

- In my original review, I stated, "On many of the datasets, such as Cliff, Gentle, Steep, Sphere, and Bilinear, the PMM performance is over 100X better than all other methods—this is shocking to me." The authors verified this interpretation, which I still find shocking. Do the authors have an explanation for the absolutely dominating performance of PMMs over all other methods? Was the training error for all approaches similarly small, and the result is about better generalization of PMMs? Or did the other methods fail to properly fit the training data?

- I remain confused why ImageNet and smaller derivatives experiments are out of reach. There are small variants such as ImageNet-100. (I think you could train e.g. a resnet50 on ImageNet for under 50 USD on cloud platforms.) I also wasn't clear why the Adam state memory made training on a larger dataset prohibitive. To be clear: not having these results is not a blocker for me to suggest acceptance, but I do think it will dampen the excitement from machine learning practitioners that benchmark datasets released over 15 years ago are beyond the scope.

- I appreciate the authors' response to my comment beginning, "The results in Table 2 ..." However, their response reinforces my view that these results are not particularly noteworthy.

- Regarding the discussion about parameter count, the authors state that, "...the number of trainable parameters directly determines the amount of training examples necessary to train any model..." This is just patently false. Results from double descent to infinite width limits and the neural tangent kernel have made abundantly clear that parameter count is not what determines the number of training examples necessary to train any model. While I do appreciate that parameter count is a good way to approximate computation cost (though again see the growth of MoEs and "active" parameters), I believe it is primarily useful when comparing computational cost *within a given model family*. The coefficient of how compute scales with parameters can vary dramatically across model types. Transformers and state-space models have very different computational profiles for the same parameter count. Moreover, neural networks are extremely quantizable and prunable (even structured pruning). Quantizing parameters to low bit precision requires less storage and pruned networks can confer major computational speedups. Again, this doesn't preclude acceptance because an imperfect measure of compute cost is better than none, but I found the statements from the authors about parameter count overly definitive.

(Remarks on code availability)

Response to Reviewer Comments on Manuscript Titled “Parametric Matrix Models”

Reviewer Comments

Reviewer #1 (Remarks to the Author):

I had reviewed this work in previous submission (Referee #1). I would like to thank the authors for this revised version of their manuscript, which addresses all of my comments.

More specifically, I appreciate the changes to the text emphasizing the kind of problems where PMMs shine, and the clarifications about the grounds of comparison with other approaches. Regarding my second point, the computational cost of PMMs, I agree with the idea that a direct comparison of CPU hours would be unfair to PMMs, although I still believe that an indicative number (with the appropriate disclaimer) would be to the benefit of the reader, as the total lack of figures might seem indicative of the prohibitive cost of this approach. However, I do appreciate the expanded section S1.1, which at least gives an idea of the computational scaling.

In conclusion, I still believe this to be an interesting approach, and the revised manuscript to be worth publishing in this new venue in its current form.

Reviewer #1 (Remarks on code availability):

I had previously reviewed the code and still believe it to be sufficiently clear and reproducible.

Reviewer #2 (Remarks to the Author):

I thank the authors for their detailed and thorough response. I am satisfied with many of the responses and believe this paper is a good fit for Nature Communications. Please assume that any points I do not follow up on below are no longer points of concern.

I am happy with the aspect of the work that focuses on PMMs as specialized to situations where “known or supposed governing equations” are given trainable parameters. I want to follow up on some of the work using PMMs as a general method.

- I still struggle with the “efficient” universal function approximator statements. How is expressivity defined in the statement that expressivity grows linearly with complexity? And what does complexity mean in this statement? I have never heard of the concept of an *efficient* universal function approximator, and if the authors are claiming something that sounds fairly strong, it should be better justified.

- I am unclear whether the hyperparameter grid search was done by the authors or by previous works. If done by the authors, I would suggest providing more information about the grid search performed. The response currently states that grid search was done over “the weight of the regularizer, the activation functions, widths of each layer, and number of hidden layers”. How many choices for each was used? More conventionally, one would not sweep over choices of activation functions and instead include the learning rate. I don’t mean to dwell on this. The more important point is that when reporting performance of other methods where the numbers do not come from extant work, it’s important to give confidence a best effort was made.

- In my original review, I stated, “On many of the datasets, such as Cliff, Gentle, Steep, Sphere, and Bilinear, the PMM performance is over 100X better than all other methods—this is shocking to me.” The authors verified this interpretation, which I still find shocking. Do the authors have an explanation for the absolutely dominating performance of PMMs over all other methods? Was the training error for all approaches similarly small, and the result is about better generalization of PMMs? Or did the other methods fail to properly fit the training data?

- I remain confused why ImageNet and smaller derivatives experiments are out of reach. There are small variants such as ImageNet-100. (I think you could train e.g. a resnet50 on ImageNet for under 50

USD on cloud platforms.) I also wasn't clear why the Adam state memory made training on a larger dataset prohibitive. To be clear: not having these results is not a blocker for me to suggest acceptance, but I do think it will dampen the excitement from machine learning practitioners that benchmark datasets released over 15 years ago are beyond the scope.

- I appreciate the authors' response to my comment beginning, "The results in Table 2 . . ." However, their response reinforces my view that these results are not particularly noteworthy.

- Regarding the discussion about parameter count, the authors state that, ". . . the number of trainable parameters directly determines the amount of training examples necessary to train any model. . ." This is just patently false. Results from double descent to infinite width limits and the neural tangent kernel have made abundantly clear that parameter count is not what determines the number of training examples necessary to train any model. While I do appreciate that parameter count is a good way to approximate computation cost (though again see the growth of MoEs and "active" parameters), I believe it is primarily useful when comparing computational cost *within a given model family*. The coefficient of how compute scales with parameters can vary dramatically across model types. Transformers and state-space models have very different computational profiles for the same parameter count. Moreover, neural networks are extremely quantizable and prunable (even structured pruning). Quantizing parameters to low bit precision requires less storage and pruned networks can confer major computational speedups. Again, this doesn't preclude acceptance because an imperfect measure of compute cost is better than none, but I found the statements from the authors about parameter count overly definitive.

Author Responses to Comments

General Comments:

We thank the reviewers for once again taking the time to give our manuscript careful consideration and provide constructive feedback. We are pleased to hear that the reviewers found our responses to their previous comments satisfactory and that they believe the manuscript is now suitable for publication in Nature Communications.

From our ongoing research on this topic, we added a statement in our introduction regarding the structure of PMMs. Specifically, we explicitly address cases in which the PMM contains only primary or only secondary matrices:

"Depending on the structure of the specific problem the PMM is being applied to, either the primary or the secondary matrices may be completely omitted—as is the case with unitary time evolution and eigenvalue emulation respectively."

Response to Reviewer #2:

I thank the authors for their detailed and thorough response. I am satisfied with many of the responses and believe this paper is a good fit for Nature Communications. Please assume that any points I do not follow up on below are no longer points of concern.

I am happy with the aspect of the work that focuses on PMMs as specialized to situations where "known or supposed governing equations" are given trainable parameters. I want to follow up on some of the work using PMMs as a general method.

*- I still struggle with the "efficient" universal function approximator statements. How is expressivity defined in the statement that expressivity grows linearly with complexity? And what does complexity mean in this statement? I have never heard of the concept of an *efficient* universal function approximator, and if the authors are claiming something that sounds fairly strong, it should be better justified.*

We thank the reviewer for catching our oversight in omitting our definition of "expressivity" and in seeking further clarification on our usage of the term "efficient". We have added an explanation of "expressivity" in the corresponding section (emphasis added to edits):

"An instructive comparison between artificial neural networks and the PMM forms considered in this

work can be made in the computational complexity of a single inference calculation as well as in the scaling of the number of trainable parameters **in relation to the ‘expressivity’ of each model. We quantify expressivity by the number of possible non-analytic points in the complex-valued output space of the model.**”

We have added explicit clarification that we are using the term “efficient” as a shorthand for models whose parameter count does not scale faster than their expressivity, and that PMMs are far from alone in this category:

“This property of expressivity scaling proportionally to complexity is how we define an ‘efficient’ universal function approximator in this work. Both the two general PMMs considered and the simplified MLP have this property.”

- I am unclear whether the hyperparameter grid search was done by the authors or by previous works. If done by the authors, I would suggest providing more information about the grid search performed. The response currently states that grid search was done over “the weight of the regularizer, the activation functions, widths of each layer, and number of hidden layers”. How many choices for each was used? More conventionally, one would not sweep over choices of activation functions and instead include the learning rate. I don’t mean to dwell on this. The more important point is that when reporting performance of other methods where the numbers do not come from extant work, it’s important to give confidence a best effort was made.

We appreciate the reviewer’s attention to detail, reproducibility, and data availability. While previously all grid search parameters could be found in the provided source code, we have now added several tables to the Supplemental Information which contain the complete grid search and data segmentation parameters for each experiment that comes from this work.

Additionally, we have greatly expanded the grid search for as many experiments as possible. This includes the Anharmonic LMG emulation, zero-error Trotter step extrapolation, and the thirteen 2D test functions in the regression experiment. These experiments now include a grid search over the learning rate for the MLP. Except in a special case described below, this did not significantly change the outcome of any experiment.

In expanding the grid search for the ALMG experiment, two mistakes in the original code were found simultaneously. The first mistake was that the choice of loss function for the grid-search cross-validation was not the mean squared error but instead `scikit-learn`’s default R^2 score (the coefficient of determination). We had previously found that the mean squared error performed best across a variety of datasets. The second mistake was that neither the primary nor secondary matrices of the PMM for this experiment were constrained to be Hermitian like in all other problems. This mistake was corrected and additionally a small improvement to the PMM was made—an insight from our ongoing research on emulating quantum systems with PMMs—in constraining the secondary matrix to be positive semi-definite just as the original observable which it aims to emulate was. By fixing these issues, the amount of training data for both methods could be cut in half, and additional analyses on the derivatives of the eigenenergies was added. The analysis of the results of this experiment now reads:

“In the right panel of Fig. 2 we show the derivative of the energy levels with respect to the control parameter ξ . Such derivatives are typically taken when searching for a phase transition in addition to, or in place of, an order parameter. We see that away from the phase transition, both models perform comparably. However, at the phase transition, only the PMM is able to accurately predict the sharp change in character of the system. This is especially prevalent in the derivatives of the energy levels.”

- In my original review, I stated, “On many of the datasets, such as Cliff, Gentle, Steep, Sphere, and Bilinear, the PMM performance is over 100X better than all other methods—this is shocking to me.” The authors verified this interpretation, which I still find shocking. Do the authors have an explanation for the absolutely dominating performance of PMMs over all other methods? Was the training error for all approaches similarly small, and the result is about better generalization of PMMs? Or did the other methods fail to properly fit the training data?

The referee asks a very good question. The key difference is that the PMM produces an interpolating function that, when continuing any variable into the complex plane with the other variables held fixed at real values, has as few non-analytic points near the real axis as possible. This leads to rapid convergence with the number of training points. The same behavior has been studied in reduced basis methods such as eigenvector continuation. In the main text of the manuscript, we have some discussion of this analysis:

“Using only general arguments of analyticity, in Ref.²⁴ it is shown that for p input features and N eigenvector snapshots, the error of the EC approximation diminishes as a decaying exponential function of $N^{1/p}$ in the limit of large N . For the case of eigenvalue problems, PMMs capture the essential features of EC calculations by proposing some unknown Hermitian matrix $M(\{c_l\})$ and then learning the matrix elements from data.”

This result follows from the exponential convergence of power series expansions with respect to expansion order for analytic functions inside their radius of convergence. Since the bilinears formed with the secondary matrices are analytic functions of the control parameters, the statement about analyticity of the interpolating function for PMMs is true for more general PMMs and not just those associated with eigenvalue problems.

For cases where the target function itself has non-analytic behavior on or very close to the real axis for some variable, the PMM will use sharp avoided level crossings in order to reproduce the non-smooth behavior. For such cases, the convergence will be slower. However, the PMM still has the advantage that the sharp avoided level crossings do not produce the unwanted oscillatory behavior typical of many other approaches. This is again explained by the statement that the PMM produces an interpolating function that has as few non-analytic points near the real axis as possible.

- I remain confused why ImageNet and smaller derivatives experiments are out of reach. There are small variants such as ImageNet-100. (I think you could train e.g. a resnet50 on ImageNet for under 50 USD on cloud platforms.) I also wasn't clear why the Adam state memory made training on a larger dataset prohibitive. To be clear: not having these results is not a blocker for me to suggest acceptance, but I do think it will dampen the excitement from machine learning practitioners that benchmark datasets released over 15 years ago are beyond the scope.

We appreciate the reviewer's input in regards to larger experiments. Our previous response could have been more clear. It is not the state memory of the optimizer but rather simply the memory of the training data that is prohibitive on the hardware available to us. These are experiments we would like to pursue in future work, as the reviewer is correct in saying that such experiments are important.

- I appreciate the authors' response to my comment beginning, "The results in Table 2 ..." However, their response reinforces my view that these results are not particularly noteworthy.

*- Regarding the discussion about parameter count, the authors state that, "... the number of trainable parameters directly determines the amount of training examples necessary to train any model..." This is just patently false. Results from double descent to infinite width limits and the neural tangent kernel have made abundantly clear that parameter count is not what determines the number of training examples necessary to train any model. While I do appreciate that parameter count is a good way to approximate computation cost (though again see the growth of MoEs and "active" parameters), I believe it is primarily useful when comparing computational cost *within a given model family*. The coefficient of how compute scales with parameters can vary dramatically across model types. Transformers and state-space models have very different computational profiles for the same parameter count. Moreover, neural networks are extremely quantizable and prunable (even structured pruning). Quantizing parameters to low bit precision requires less storage and pruned networks can confer major computational speedups. Again, this doesn't preclude acceptance because an imperfect measure of compute cost is better than none, but I found the statements from the authors about parameter count overly definitive.*

The reviewer draws attention to a critical overloading of terminology here. In the field of machine learning, “train” or “fit” usually means optimizing a model’s performance on data. However, there are many contexts in scientific emulation where these terms are about determining the model parameters themselves. It is absolutely true that if the goal is only to optimize a model’s performance, then the number of trainable parameters does not determine the amount of data required. If instead the goal is the determination of the unknown parameters, then the number of such parameters does indeed determine the amount of data necessary. The machine learning definitions are likely the more immediately recognized ones, and as such we have carefully rewritten the section to read:

“In this work we consider the number of trainable parameters as a concise and interpretable indicator of efficiency. This is not because the limiting factor in machine learning is the storage of model parameters. Instead it is because—for the forms of PMMs in this work and the ubiquitous feedforward neural network—training and inference complexity scales proportionally to the number of trainable parameters. This relation also holds for individual layers of significantly more advanced neural network architectures such as the self-attention layer which composes the transformer deep learning architecture, and which in turn plays a central role in Large Language Models.²⁸ This is discussed further in Methods. Additionally in the context of emulating physical systems, where the trainable parameters may have physical significance and therefore the determination of the parameters is potentially as important as finding a model that reproduces the data, the number of trainable parameters provides a lower bound on the number of informative training examples necessary to identify sufficiently narrow neighborhoods of parameter solutions.²⁹”

This final sentence is true so long as the model is well-structured and is therefore able to fit the data well—a reasonable assumption in the context of sufficiently expressive universal function approximators.

We agree with the reviewer that the coefficient of how compute scales with parameters can be modified through various techniques and changes between model types. We have focused on the “big O” scaling in this work to provide an instructive—but not authoritative—indication of the computational cost independent of these prefactors. We hope to explore some of the methods the reviewer has mentioned to provide large constant-factor improvements in future work.